# A Syntenin Inhibitor Blocks Endosomal Entry of SARS-CoV-2 and a Panel of RNA Viruses

**DOI:** 10.3390/v14102202

**Published:** 2022-10-06

**Authors:** Richard Lindqvist, Caroline Benz, Vita Sereikaite, Lars Maassen, Louise Laursen, Per Jemth, Kristian Strømgaard, Ylva Ivarsson, Anna K. Överby

**Affiliations:** 1Department of Clinical Microbiology, Umeå University, 90185 Umeå, Sweden; 2Laboratory for Molecular Infection Medicine Sweden (MIMS), Umeå University, 90186 Umeå, Sweden; 3Department of Chemistry—BMC, Uppsala University, Box 576, Husargatan 3, 75123 Uppsala, Sweden; 4Center for Biopharmaceuticals, Department of Drug Design and Pharmacology, University of Copenhagen, Universitetsparken 2, 2100 Copenhagen, Denmark; 5Department of Medical Biochemistry and Microbiology, Uppsala University, Box 582, Husargatan 3, 75123 Uppsala, Sweden

**Keywords:** SARS-CoV-2, CHIKV, flavivirus, syntenin, peptide inhibitor

## Abstract

Viruses are dependent on host factors in order to efficiently establish an infection and replicate. Targeting the interactions of such host factors provides an attractive strategy to develop novel antivirals. Syntenin is a protein known to regulate the architecture of cellular membranes by its involvement in protein trafficking and has previously been shown to be important for human papilloma virus (HPV) infection. Here, we show that a highly potent and metabolically stable peptide inhibitor that binds to the PDZ1 domain of syntenin inhibits severe acute respiratory syndrome coronavirus 2 (SARS-CoV-2) infection by blocking the endosomal entry of the virus. Furthermore, we found that the inhibitor also hampered chikungunya infection and strongly reduced flavivirus infection, which is completely dependent on receptor-mediated endocytosis for their entry. In conclusion, we have identified a novel broad spectrum antiviral inhibitor that efficiently targets a broad range of RNA viruses.

## 1. Introduction

Viruses are obligate intracellular parasites that depend on interactions with host proteins and employ the cellular machinery of the host for all stages of the viral life cycle, from viral entry to exit. SARS-CoV-2 is, for example, known to enter host cells by two different mechanisms (I): endosomal entry in the absence of transmembrane serine protease 2 (TMPRSS2) or (II) direct fusion at the plasma membrane in the presence of TMPRSS2 [1]. Among the host proteins exploited by pathogenic viruses are members of the PSD95/DLG/ZO-1 (PDZ) protein domain family [2,3]. PDZ domains typically interact with their peptide-binding partners through short sequences found at the C-terminus of target proteins. PDZ mediated interactions are crucial for several cellular processes, including the clustering of transmembrane proteins and trafficking. For example, the PDZ domain-containing, protein-sorting nexin 27 (SNX27) is involved in the retrograde transport from endosome to plasma membrane [4] and binds among other ligands to the C-terminal TxF-COO- motif (x indicates any amino acid) of the angiotensin-converting enzyme 2 (ACE2) [5], which SARS-CoV-2 virus particles use a receptor to enter cells. Recently, it was shown that SNX27 directs the trafficking of the complex between ACE2 and SARS-CoV-2 virus and prevents its lysosome/late endosome entry [6].

Here, we focus on the PDZ-containing protein syntenin. Syntenin is involved in clustering and trafficking of transmembrane receptors [7], such as the heparan sulfate proteoglycans syndecan1–4 [8] and the tetraspanin CD63 [9], as well in the biogenesis of exosomes [10]. Among the syntenin ligands, syndecans have been proposed to facilitate SARS-CoV-2 cell attachment [11] and viral uptake [12,13]. Syntenin has also been shown to bind to the C-terminal PDZ-binding motif of the envelope (E) protein of SARS-CoV [14]. Although truncation of the PDZ binding motif of the SARS-CoV E protein did not affect the viral growth in a murine infection model, the mutated virus caused less lung damage and mortality than the wild-type virus. The E protein of SARS-CoV and SARS-CoV-2 are further recognized by a set of other PDZ proteins, including MPP5 (also called INADL) [15,16]. In addition to its role in assembly, budding and virulence, the E protein serves an important role during SARS-CoV-2 entry, as the E protein fuses with the endosomal or plasma membrane in order to release the viral RNA into the cytoplasm [17,18]. The SARS-CoV-2 ORF3a also has a PDZ-binding motif, shown to bind to other PDZ proteins [19]. During infection, ORF3a acts as a viral pathogenicity factor by inducing cytokine storm and also regulate apoptosis [20,21]. In addition to the reported PDZ-binding motifs, there is a putative motif in the C-terminal region of the short intrinsically disordered NSP11 (full sequence SADAQSFLNGFAV-COOH). NSP11 becomes the N-terminus of NSP12 in the ribosomal frameshift of ORF1b. However, the function of NSP11 is not fully understood [22]. Thus, several lines of evidence implicate potential roles of syntenin in SARS-CoV-2 infection and suggest that inhibiting syntenin could be a valid antiviral strategy.

We recently described KSL-128114, a highly potent and metabolically stable cell-penetrating peptide inhibitor that bind to the first (PDZ1) of two PDZ domains of syntenin [23]. We reasoned that the KSL-128114 inhibitor could potentially be used to affect syntenin’s role in receptor trafficking, as well as potential interactions between syntenin and the SARS-CoV-2 proteins, which could affect viral infection. Here, we first explored to what extent the PDZ1-2 tandem of syntenin binds to the known and putative PDZ-binding motifs of SARS-CoV-2 proteins. We then demonstrated an antiviral effect of the PDZ inhibitor KSL-128114 to the blockage of SARS-CoV-2 entry, but not membrane fusion dependent entry. Furthermore, we found that KSL-128114 inhibited chikungunya virus (CHIKV) and several flaviviruses, which are dependent on endosomal entry [24]. Thus, we demonstrated that the syntenin inhibitor KSL-128114 can be used as a broad spectrum inhibitor of viral infection.

## 2. Materials and Methods

### 2.1. Expression and Purification of Proteins

*E. coli* BL21(DE3) gold bacteria (Agilent Technology) containing pETM33 plasmids encoding the 6-His-GST fusion proteins (human Syntenin PDZ1-2 (amino acids 111-213), MPP5 PDZ (amino acids 238–336) and SNX27 PDZ (amino acids 740–141); synthetic genes obtained from GeneScript)) were grown in 4 L 2xYT (16 mg/mL peptone, 10 mg/mL yeast extract, 5 mg/mL NaCl) at 37 °C in a rotary shaker (200 rotations per minute). For each protein, expression was induced with 1 mM isopropyl β-D-1-thiogalactopyranoside (IPTG) and was allowed to proceed for 16 h at 18 °C. Bacteria were harvested for 10 min at 4500 g. The bacterial pellet was resuspended in lysis buffer (7.5 mL PBS supplemented with 1% Triton, 10 µg/mL DNaseI, EDTA-free Protease Inhibitor Cocktail (Roche)) and was incubated for 1 h on ice. The suspension was sonicated to destroy remaining DNA and support the lysis, and the cell debris was pelleted by centrifugation (1 h, 16,000 g). Proteins were batch purified from the supernatant using Ni Sepharose^®^ Excel (Cytiva) using the manufacturer’s recommended buffers. The supernatant was mixed with the matrix and unbound fraction was washed out with wash buffer (20 mM NaPi, 0.5 M NaCl, 30 mM imidazole, pH 7.4). The bound protein was cleaved with His-tagged 3C protease (in 20 mM NaPi, 0.5 M NaCl, pH 7.4) at 4 °C for 16 h. The proteolytically released PDZ domains were obtained from the matrix by addition of buffer. The protein size and purity were confirmed through SDS-PAGE. Purified proteins were dialyzed into 50 mM potassium phosphate buffer, pH 7.5, for 16 h.

### 2.2. Peptide Synthesis

KSL-128114 was synthesized using standard Fmoc-based SPPS on a Prelude X, induction heating-assisted, peptide synthesizer (Gyros Protein Technologies, Tucson, AZ, USA) with 10 mL glass reaction vessel using preloaded Wang-resins (100–200 mesh). Reagents were prepared as solutions in DMF: Fmoc-protected aa (0.2 M), O-(1H-6-chlorobenzotriazole-1-yl)-1,1,3,3- tetramethyluronium hexafluorophosphate (HCTU, 0.4 M) and N,N-diisopropylethylamine (DIPEA) (0.8 M). Sequence elongation was achieved using the following protocol: deprotection (2 × 2 min, rt, 350 rpm shaking) and coupling (2 × 5 min, 50 °C, 350 rpm shaking). Amino acids were double coupled using amino acid/HCTU/DIPEA (ratio 1:1:2) in 5-fold excess over the resin loading to ensure efficient peptide elongation.

N-terminal labeling of KSL-128114 with 5-(and-6)-carboxytetramethylrhodamine (TAMRA, Anaspec Inc.) was performed on resin, by coupling TAMRA for 16 h at rt, using a mixture in NMP of TAMRA: (benzotriazol-1-yloxy) tripyrrolidinophosphonium hexafluorophosphate (PyBOP):DIPEA (1.5:1.5:3). To avoid photobleaching of the fluorophore, the reaction vessel was covered and the coupling finalized with extensive resin washes with DMF and DCM.

### 2.3. Peptide Cleavage and Purification

The synthesized peptides were cleaved from the resin using a mixture of 90% TFA, 2.5% H_2_O, 2.5% TIPS, 2.5% 1,2-ethanedithiol (EDT) and 2.5% thioanisole. After cleavage, the peptides were precipitated with ice-cold diethyl ether and centrifuged at 2500× *g* for 10 min at 4 °C. The resulting peptide precipitate was redissolved in 50:50:0.1 (H_2_O:MeCN:TFA) and lyophilized. Purification of all peptides were performed with a preparative reverse phase high performance liquid chromatography (RP-HPLC) system (Waters) equipped with a reverse phase C18 column (Zorbax, 300 SB-C18, 21.2 × 250 mm) and using a linear gradient with a binary buffer system of H2O:MeCN:TFA (A, 95:5:0.1; B, 5:95:0.1) (flow rate 20 mL/min). The collected fractions were characterized by LC-MS. The purity of the fractions was determined at 214 nm on RP-UPLC.

### 2.4. Fluorescence Polarization

Affinity measurements were carried out using fluorescence polarization in an iD5 multi detection plate reader (Molecular Devices) using Corning assay 96 well half area black flat-bottom non-binding surface plates (Corning, USA #3993). The settings were 485 nm excitation and 535 nm for emission at a reading height of 1.76 mm and total volume of 50 µL. Peptides were obtained from GeneCust (France) at >95% purity. Unlabeled peptides were dissolved in 50 mM potassium phosphate, pH 7.5. Fluorescein isothiocyanate (FITC)-labeled peptides were dissolved in dimethyl sulfoxide (DMSO). Protein for saturation experiments, or peptides for the displacement experiments, were arrayed in serial dilution in 50 mM potassium phosphate, pH 7.5 in 25 µL, followed by addition of 25 µL of a master mix. In case of saturation-binding experiments, the master mix contained 2 mM DTT and 10 nM FITC-labeled peptide in 50 mM potassium phosphate, pH 7.5. For competition experiments, the master mix was supplemented with the protein of interest at a concentration of 4 times the K_D_ value.

### 2.5. Cells and Viruses

VeroE6 cells were cultured in DMEM (Sigma), containing 5% fetal bovine serum (FBS), 100 U/mL of penicillin and 100 µg/mL streptomycin (Gibco). VeroB4 cells were grown in medium 199/EBSS (Hyclone), supplemented with 10% FBS (Hyclone), 100 U/mL of penicillin and 100 μg/mL streptomycin (Gibco). Calu-3 and HEK293T hACE2 cells were grown in DMEM (Sigma), supplemented with 10% FBS (Hyclone), 100 U/mL of penicillin and 100 μg/mL streptomycin (Gibco). The patient isolate SARS-CoV-2/01/human/2020/SWE accession no/GeneBank no MT093571.1 was provided by the Public Health Agency of Sweden. The virus was passaged 4 times in VeroE6. Tick-borne encephalitis virus (TBEV) (Torö-2003, infectious clone 2 passages in VeroB4 cells [25], West Nile virus (WNV) (isolated in 2003 in Israel WNV_0304h_ISR00, passage number 5) and dengue virus (DENV)-2 (PNG/New Guinea C). WNV and DENV were kind gifts from Dr. S. Vene (Public Health Agency of Sweden, Stockholm, Sweden). CHIKV (CHIKV LR2006OPY1) was a kind gift of Magnus Evander (Umeå University, Umeå, Sweden). DENV, WNV, TBEV and CHIKV were grown and titrated in VeroB4 cells.

### 2.6. Viral Infections

VeroE6 and Calu-3 cells were infected with SARS-CoV-2 (MOI: 0.05) and VeroB4 cells were infected with either DENV, WNV, TBEV (MOI: 0.1) or CHIKV (MOI: 0.05) for 1 h at 37 °C and 5% CO_2_. Then, inoculum was removed and replaced with medium containing either the indicated amount of KSL-128114, Chloroquine (Sigma, C6628) or DMSO. After 16 h (SARS-CoV-2 and CHIKV) or 24 h (DENV, WNV and TBEV) of infection, cells were fixed in 4% formaldehyde for 30 min, permeabilized in PBS, 0.5% trition-X-100 and 20 mM glycine. Virus was detected using primary monoclonal rabbit antibodies directed against SARS-CoV-2 nucleocapsid (Sino Biological Inc., 40143-R001), or monoclonal mouse antibodies directed against Flavivirus E protein (HB112 ATCC), or monoclonal mouse antibodies directed against TBEV E [26] and conjugated secondary antibodies anti-rabbit Alexa555 (1:500, Thermo Fisher Scientific). Nuclei were counterstained with DAPI. Number of infected cells were quantified using a TROPHOS plate RUNNER HD. The infection was normalized to the number of nuclei and presented as % infection, compared to DMSO control. For binding and entry assays, cells were first pre-treated using 30 µM KSL-128114 for 2 h, and then infected with MOI:1 using ice-cold medium, containing either 30 µM KSL-128114 or DMSO at 4 °C. To analyze binding, inoculum was removed after 1 h of infection and cells were washed 3 times with PBS and then lysed. To analyze entry of virions into cells, inoculum was removed after 1 h of infection and replaced with fresh medium. Cells were then incubated for another 2 h at 37 °C and 5% CO_2_ before being washed with PBS-EDTA, trypsinized for 10 min, washed 3 times in PBS, and then lysed. Viral burdens were measured using qPCR.

### 2.7. Time of Addition Assay

VeroE6 cells were treated with 30 μM KSL-128114 according to the following setup, (I, “−2”): cells were treated with 30 μM KSL-128114 for 2 h at 37 °C and 5% CO_2,_ then medium containing peptide was removed and cell were infected with SARS-CoV-2 (MOI: 0.05) for 1 h 37 °C and 5% CO_2_, then inoculum was replaced with fresh medium, containing 30 μM KSL-128114 and cells were incubated at 37 °C and 5% CO_2_. (II, “1”): Cells were infected with SARS-CoV-2 (MOI:0.05) for 1 h at 37 °C and 5% CO_2_, then inoculum was replaced with fresh medium, containing 30 μM KSL-128114 and incubated at 37 °C and 5% CO_2_. (III, “3”) Cells were infected with SARS-CoV-2 (MOI:0.05) for 1 h at 37 °C and 5% CO_2_, then inoculum was replaced with fresh medium, after 2 h medium was replaced with fresh medium, containing 30 μM KSL-128114, and cells were incubated at 37 °C and 5% CO_2_. After 16 h of infection, cells were fixed using 4% formaldehyde and permeabilized in 0.5% triton-X 100, 20 mM glycine in PBS. Infected cells were detected using primary monoclonal rabbit antibodies directed against SARS-CoV-2 nucleocapsid (Sino Biological Inc., 40143-R001) and conjugated secondary antibodies anti-rabbit Alexa555 (1:500, Thermo Fisher Scientific). Nuclei were counterstained with DAPI.

### 2.8. Cell Viability Assay and qPCR

Cellular viability was measured using Cell Titer Glo (Promega), according to the manufacturer’s instructions. Luminescence was measured on a TECAN infinite F200PRO plate reader. Viral RNA from supernatants were isolated from 100 µL supernatant using the QIAamp Viral RNA Mini Kit (Qiagen) and viral RNA from cell lysate was extracted using the Nucleospin RNA plus mini kit (Macherey-Nagel), according to the manufacturer’s instructions. Ten microliters of RNA were used to synthesize cDNA using High Capacity cDNA Reverse Transcription Kit (Applied Biosystems) according to the manufacturer’s instructions. GAPDH transcripts were detected by RT2 qPCR Primer Assay (Qiagen, Cat# 330001 PPQ00249A) and the qPCRBIO SyGreen Mix Hi-ROX kit (PCRBIOSYSTEMS); viral transcripts were detected using the qPCRBIO Probe Mix Hi-ROX kit (PCRBIOSYSTEMS) and the indicated primers and probes (Table 1). For strand specific qPCR the random primers of High Capacity cDNA Reverse Transcription Kit were replaced with either the forward or reverse SARS-CoV-2 primer. The qPCR was run on a StepOnePlus fast real-time PCR system (Applied Biosystems).

### 2.9. Immunofluorescence Stainings

HEK293T hACE2 and VeroE6 cells were treated with 30 μM KSL-128114 or DMSO for 6 h, and then fixed in 4% formaldehyde. Cells were then either permeabilized in 0.5% Triton X-100 or left unpermeabilized. Then, cellular expression of ACE2 or syndecan-1 were detected by rabbit monoclonal antibodies (ACE2, Novus bio, NBP2-67692, 1:500, Syndecan-1, abcam, ab128936 2 μg/mL) and secondary donkey anti-rabbit Alexa Fluor 488 (1:2000, Thermo Fisher Scientific). Cellular fluorescence was measured using a TROPHOS plate RUNNER HD.

## 3. Results

### 3.1. Syntenin Binds with Low Affinity to the SARS-CoV-2 E Protein and the SARS-CoV-2 NSP11

We obtained synthetic FITC-labeled peptides, corresponding to the C-termini of the E protein, ORF3 and NSP11, respectively, from SARS-CoV-2 (Figure 1A). These peptides were used to determine the affinities for recombinantly expressed and purified syntenin PDZ1-PDZ2, SNX27 PDZ and MPP5 PDZ. The affinity determinations (Figure 1B) revealed that syntenin PDZ1-2 binds with the highest affinities to the putative PDZ-binding motif found at the C-terminus of NSP11_4393-4405_ (KD = 133 ± 25 μM) and with lower affinity to the peptide from the E protein (K_D_ not determinable). The affinities are low but comparable to endogenous syntenin interactions [23]. The interaction with oligomeric E protein may be enhanced by avidity effects in a cellular setting [32]. MPP5 PDZ, that was added as a control, bound, as previously reported, preferentially to the E protein (K_D_ = 400 ± 55 µM). In contrast, SNX27 bound all three peptides with low affinity (K_D_ > 400 μM).

### 3.2. KSL-128114 Inhibits Viral Infection

Having confirmed that syntenin can interact with NSP11, and to a lesser extent with E (Figure 1B), we aimed to explore the consequences on viral infection and replication by inhibiting the interactions with syntenin using the cell-penetrating peptide-based inhibitor KSL-128114. We, therefore, determined the level of SARS-CoV-2 infection as a function of inhibitor concentration in VeroE6 cells and found that KSL-128114 efficiently inhibited viral infection (EC50 = 20 μM) with zero or minor effects on cell viability. By treating the cells with 30 μM inhibitor, we found that both viral infection and release of new viral particles were reduced (Figure 2B,C). To gain more insight into the antiviral mechanism of the inhibitor, we performed an experiment, where the time of addition of inhibitor was investigated. We evaluated the effect of adding the inhibitor 2 h before and 1 or 3 h after infection (Figure 2D). Although treatment prior to infection had a striking effect on the infection level, post-infection treatment had no effect, suggesting that the inhibitor blocks the early steps of viral infection, taking place prior to any interactions between syntenin and viral proteins.

### 3.3. KSL-128114 Blocks SARS-CoV-2 Entry into Cells

Intriguingly, the observed antiviral effect of the syntenin inhibitor could not be caused by blocking the interactions between the intracellular PDZ proteins and the viral PDZ-binding motifs, as these interactions would occur at a later stage during infection. The results instead suggested that the inhibitor blocks important endogenous interactions needed early in the viral life cycle, for example between ACE2 and PDZ proteins involved in ACE2 endocytosis and recycling. We reasoned that the inhibitor could bind off-target to the PDZ domain of SNX27. It was recently found through a genome-wide CRISPR screen that knock out of SNX27 and other components of the retromer complex involved in the recycle of the receptors back to the plasma membrane inhibits viral replication [33]. We tested if the inhibitor could bind to SNX27 and found that it bound with 15-fold lower affinity to SNX27 in comparison to the target syntenin (K_D_ = 0.3 ± 0.18 μM for syntenin; K_D_ = 5.0 ± 0.2 μM for SNX27, Figure 3A), making it unlikely that this would be the main way to explain the antiviral effect. We further reasoned that syntenin PDZ1-2 could potentially bind to ACE2 C-terminus and be directly involved in its trafficking. However, syntenin did not bind to ACE2 in our FP-based affinity measurement (Figure 3B), and treatment of HEK293T hACE2 cells with the syntenin inhibitor did not alter the cell surface expression of ACE2 (Figure 3C). However, we found that the inhibitor increased the total amount of ACE2 expression; the most likely explanation is related to off-target effects of the inhibitor on a panel of other PDZ domains [5,23]. Instead, we found that inhibitor treatment led to a lower expression level of the known syntenin cargo syndecan-1 on the surface of VeroE6 cells (Figure 3D). Consistent with previously reported data, the results indicated that the inhibitor confers a block of the syntenin-dependent endocytic trafficking [23].

Next, we examined if the inhibitor affects binding of the virus to the host cell or the viral entry into cells by treating the cells with the inhibitor 2 h before infection and measuring viral RNA with real time qPCR after 1 h on ice (binding) or after 2 h infection (entry). We detected no change in SARS-CoV-2 binding to cells in the presence of the inhibitor (Figure 3E). Finally, we investigated if the inhibitor affected the entry of virus into cells. To this end, we removed virus particles that were bound to the cells but had not entered by trypsination and measured the levels of positive and negative stranded ssRNA separately. Whereas the presence of positive strand ssRNA is an indicator for the viral genome, the negative strand will only be detected if the viral genome has entered the cytoplasm, initiated protein translation and started to replicate. We found that the levels of both positive- and negative-stranded ssRNA were reduced (Figure 3F), thus, indicating inhibition of viral entry.

TMPRSS2 is largely absent in VeroE6 cells forcing the virus to enter by the endosomal pathway. In contrast, the lung epithelial cell line Calu-3 expresses TMPRSS2. We hypothesized that if KSL-128114 specifically targets the endosomal pathway then the inhibitory effect of KSL-128114 on SARS-CoV-2 infection would be reduced in Calu-3 cells. Consistent with the hypothesis, KSL-128114 had no inhibitory or toxic effect on SARS-CoV-2 infection in Calu-3 cells (Figure 3G,H). Similar findings have been shown with the inhibitor chloroquine, which prevents the acidification of the endosomes. Chloroquine has been shown to inhibit SARS-CoV-2 infection in the absence of TMPRSS2 but have less effect on infection in the presence of TMPRSS2 [34]. To further investigate the role of TMPRSS2 in Calu-3 cells, we treated infected Calu-3 and VeroE6 cells with chloroquine and monitored the infection (Figure 3I) and viability (Figure 3J). We found that viral infection was strongly inhibited by blocking endosomal entry with chloroquine in VeroE6, lacking TMPRSS2 expression. However, in Calu-3 cells the TMPRSS2 expression rendered the cells resistant to chloroquine treatment. The results with the syntenin inhibitor, thus, follow the same line as the results with chloroquine treatment. Taken together, we showed that the syntenin inhibitor, KSL-128114, is a novel endosomal entry inhibitor of SARS-CoV-2 infection.

### 3.4. The Syntenin Inhibitor Can Be Used as a Broad Spectrum Antiviral Agent

Many enveloped viruses utilize the endosomal trafficking for their uptake. We reasoned that syntenin might be assisting the post-endocytic step of the uptake of other viruses as well and that the inhibitor, thus, could be applicable to a broader panel of viruses. We, therefore, tested the effect of the syntenin inhibitor on infection by a set of enveloped ssRNA viruses, namely a panel of flaviviruses (dengue virus (DENV), West Nile virus (WNV) and tick-born encephalitis virus (TBEV)) and an alphavirus (CHIKV). Consistent with the hypothesis, we found that treatment with KSL-128114 strongly inhibited viral infection of both flaviviruses and alphavirus (Figure 4A,C) and the release of progeny virus (Figure 4B,D). Thus, KSL-128114 is a novel pan-viral entry inhibitor that acts by blocking the endosomal entry pathway.

## 4. Discussion

As obligate intracellular parasites viruses need to hijack cellular proteins in order to establish an infection and carry out their life cycle. Syntenin is a protein involved in the trafficking of proteins, including proteins important for SARS-CoV-2 attachment and uptake [11,12,13], which makes it an interesting target for antiviral therapies. KSL-128114 is a highly potent cell-penetrating peptide inhibitor of syntenin and might disrupt syntenin’s interaction with viral proteins and/or host factors targeted by viruses. To investigate the potential of KSL-128114 as an antiviral we treated cells with the peptide inhibitor [23] and found it to inhibit SARS-CoV-2 infection, as well as the infection of several RNA viruses.

On closer investigation, we found that KSL-128114 did not block the binding of the SARS-CoV-2 virus to the VeroE6 cells, but rather the post-endocytic entry of the virus to the cytoplasm. The inhibitor appears to specifically block SARS-CoV-2 entry by the endosomal pathway, as it was highly effective on VeroE6 cells that lack TMPRSS2 but failed to inhibit infection of Calu-3 cells that expresses TMPRSS2, which enables SARS-CoV-2 to fuse at the plasma membrane [35]. In that sense, KSL-128114 showed similar inhibition patterns as chloroquine, which strongly inhibited SARS-CoV-2 infection in VeroE6 cells but not in Calu-3 cells. Chloroquine acts by increasing the pH within the endosomes, and thus inhibits viruses that depend on low pH for their entry [36]. Previous studies have indeed showed similar findings, that chloroquine inhibited SARS-CoV-2 in VeroE6 but not in Calu-3 cells [34,37], indicating that the entry mechanism in VeroE6 cells are dependent on the endosomal pathway. The KSL-128114 inhibitor has previously been shown to have a negative effect on syntenin-dependent endosomal budding through binding to syntenin PDZ1 and has been shown to block an interaction between syntenin and RAB5 [23], a key factor in regulating early endocytosis [38,39]. The impairment of the PDZ1-binding pocket has further previously been shown to lead the co-accumulation of syntenin and syndecan in a recycling compartment [8]. As syndecans have been implicated in facilitating SARS-CoV-2 viral uptake [12,13], it is plausible that inhibited syntenin-dependent trafficking of syndecans blocked the virus from endosomal escape.

However, syntenin is involved in trafficking of many other transmembrane proteins, including the tetraspanin CD63, and the exact mechanism of the inhibition of endosomal escape, thus, remains unclear. Notably, the CD63-syntenin complex has been found to be involved in post-endocytic trafficking of human papillomavirus (HPV). In this case, the internalized viral particles are transported to multivesicular endosomes, where acidification and disassembly occur in a CD63-syntenin-Alix dependent process [40]. Syntenin has also been shown to interact with other tetraspanins, such as CD9 and CD81 [9], and CD9 has been shown to facilitate the entry of the closely related coronavirus, Middle East Respiratory Syndrome coronavirus (MERS-CoV) [41]. We found that the inhibitor could reduce CHIKV infection, and CD9 is also implicated in efficient CHIKV entry and infection [42,43]. In addition, there are links between tetraspanins and virus-mediated vesicular trafficking of enveloped viruses, including flaviviruses [44]. Furthermore, KSL128114 was shown to disrupt the interaction between Rab5 and syntenin [23], and Rab5 is needed for DENV and WNV entry in to cells [45], which may suggest an interesting mechanism for KSL-128114 inhibition of viral entry.

Finally, we confirmed that syntenin can interact with low affinity with the E protein of SARS-CoV-2, a protein that is involved in membrane fusion at the endosomal membrane [17]. If the interaction between syntenin and the E protein is needed to facilitate viral entry, the inhibitor could also contribute to the inhibition of viral entry by disrupting this interaction.

Whether KSL-128114 blocks trafficking of syndecans, CD63, other syntenin cargos or direct interactions between syntenin and with viral proteins to confer the antiviral effect remain to be elucidated and may vary from virus to virus. Nevertheless, our results clearly demonstrates that the treatment of cells with a syntenin inhibitor can be used to inhibit infection of a broad range of enveloped RNA viruses, most likely by blocking the entry through the endosomal pathway, which might suggest a path towards the development of novel antiviral therapeutics.

## 5. Conclusions

In this study, we demonstrate the importance of syntenin for infection of a wide-range of different RNA viruses. By treating cells with a highly potent and metabolically stable peptide inhibitor of syntenin, we could inhibit the infection of several viruses, such as SARS-CoV-2, CHIKV, DENV, WNV and TBEV. We found that the inhibitor was acting on the very early stages of viral infection, most likely the entry step. Interestingly, the inhibitor could only inhibit viral infection on cells and viruses, utilizing the endosomal entry pathway, and failed to inhibit SARS-CoV-2 entry via the plasma membrane. This indicates that syntenin is needed for receptor-mediated endocytosis. Taken together, we found that a peptide inhibitor of syntenin can be used as broad spectrum viral entry inhibitor.

## Figures and Tables

**Figure 1 viruses-14-02202-f001:**
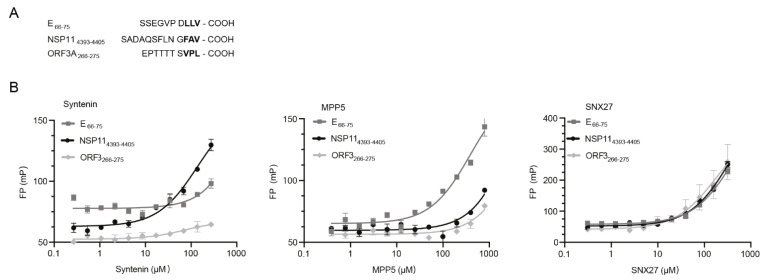
PDZ-binding motifs in SARS-CoV-2 proteins. (**A**) Amino acid sequences of the known (E66-77 and ORF3A_266-275_) and putative (NSP11_4393-4405_, numbering according to polyprotein). (**B**) FP-based affinity determinations of syntenin PDZ1-2 (left), MPP5 PDZ (middle) and SNX27 PDZ (right) and FITC-labeled peptides derived from the C-termini of SARS-CoV-2 E, ORF3 and NSP11, respectively. The K_D_ values of syntenin PDZ1-2-binding NSP11 (133 ± 25 µM) and MPP5-binding E (400 ± 40 µM) could be calculated. Other interactions were occurring with apparently lower affinity.

**Figure 2 viruses-14-02202-f002:**
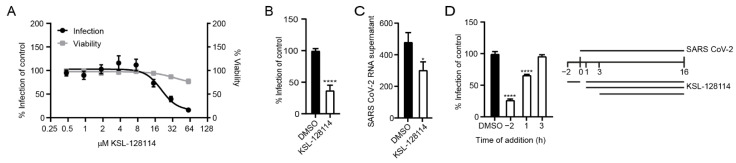
KSL-128114 inhibits SARS-CoV-2 infection. (**A**) Dose–response curve of KSL-128114. VeroE6 cells were treated with KSL-128114 and were infected with SARS-CoV-2 for 16 h. Inhibition of viral infection was quantified using a TROPHOS plate RUNNER HD, and toxicity of KSL-128114 was monitored using Cell Titer Glo (Promega) (N = 9). (**B**,**C**) Cells were treated with 30 μM KSL-128114 and infected with SARS-CoV-2 for 16 h. Number of infected cells were quantified using a TROPHOS plate RUNNER HD (N = 18) and viral RNA in the supernatants was detected using qPCR (N = 6). (**D**) Time of addition dependence of KSL-128114 inhibition. VeroE6 cells were treated 2 h before, or 1 or 3 h post-SARS-CoV-2 infection. Infection was quantified using a TROPHOS plate RUNNER HD (N = 6). All experiments were performed in at least two independent experiments. Statistical significance was calculated by unpaired t test using GraphPad Prism. Asterisks indicate statistical significance, * *p* < 0.05, **** *p* < 0.0001.

**Figure 3 viruses-14-02202-f003:**
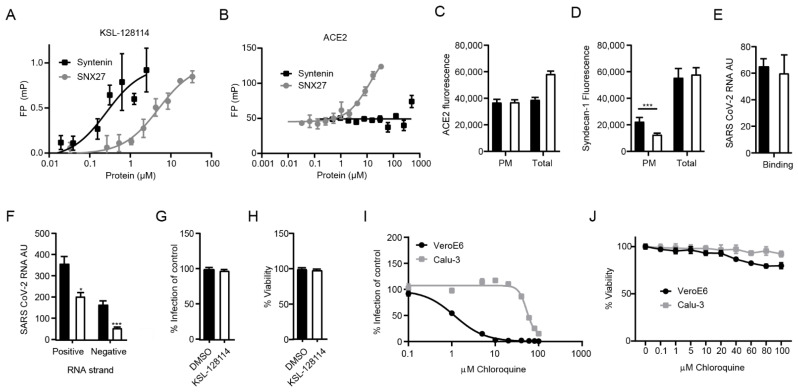
KSL-128114 inhibits SARS-CoV-2 entry of cells. (**A**) FP-based affinity measurement of 5(6)-carboxytetramethylrhodamine (TAMRA) labeled KSL-128114 binding to syntenin PDZ1-2 or SNX27 PDZ. There was a more than 15-fold difference in affinity for the proteins for the inhibitor. (**B**) FP-based affinity measurement of FITC-labeled ACE2 peptide (aa start-end) to syntenin and SNX27 PDZ domains. Note that syntenin PDZ1-2 do not bind ACE2. (**C**) Detection of ACE2; cells were permeabilized in order to detect ACE2 on the cellular surface, as well as inside the cell “total”, or cells were not permeabilized in order to detect ACE2 on the cellular surface “PM” (plasma membrane). ACE2 in HEK293 hACE2 cells in the presence of 30 μM KSL-128114 (white bars) or DMSO (black bars) (N = 6). (**D**) Detection of total syndecan-1 and plasma membrane bound syndecan-1 in VeroE6 cells in the presence of 30 μM KSL-128114 or DMSO (N = 6). (**E**) Binding assay: VeroE6 cells were infected (MOI: 1) for 1 h at 4 °C, cells were washed and lysed and viral RNA was measured by qPCR (N = 6). (**F**) Entry assay: Cells were infected for 1 h (MOI: 1), washed and incubated for 2 h at 37 °C. Bound but not entered virus was removed by trypsin and cells were lysed, and strand-specific viral RNA was detected by qPCR (N = 6). (**G**) Antiviral assay Calu-3 cells, cells were treated with 30 μM KSL-128114 and infected with SARS-CoV-2 for 16 h, number of infected cells were quantified using a TROPHOS plate RUNNER HD (N = 9). (**H**) Viability of uninfected, KSL-128114 treated Calu-3 cells, cellular viability was measured using Cell Titer Glo (N = 9). (**I**) VeroE6 and Calu-3 cells were treated with the indicated concentration of chloroquine and number of infected cells were quantified using a TROPHOS plate RUNNER HD (N = 6). (**J**) Viability of uninfected cells, VeroE6 and Calu-3 cells were treated with the indicated concentration of chloroquine and viability was measured using Cell Titer Glo (N = 6). All experiments were performed in at least two independent experiments. Statistical significance was calculated by unpaired t test using GraphPad Prism. Asterisks indicate statistical significance, * *p* < 0.05, *** *p* < 0.001.

**Figure 4 viruses-14-02202-f004:**
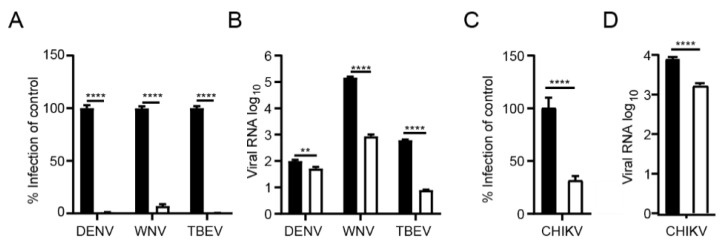
KSL-128114 inhibits flaviviruses and alphavirus. (**A**,**B**) Cells were treated with 30 μM KSL-128114 and infected with DENV, WNV and TBEV for 24 h (MOI: 0.1), number of infected cells were quantified using a TROPHOS plate RUNNER HD and viral RNA in the supernatants was detected using qPCR. (**C**,**D**) Cells were treated with 30 μM KSL-128114 and infected with CHIKV for 16 h (MOI: 0.05), number of infected cells were quantified using a TROPHOS plate RUNNER HD and viral RNA in the supernatants was detected using qPCR. Experiments were performed twice in 5 replicates (N = 10). Statistical significance was calculated by unpaired t test using GraphPad Prism. Asterisks indicate statistical significance, ** *p* < 0.01, **** *p* < 0.0001.

**Table 1 viruses-14-02202-t001:** Primers and probes used to detect viral RNA.

Target	Sequence	Reference
SARS-CoV-2 forward primer	GTCATGTGTGGCGGTTCACT	[27]
SARS-CoV-2 reverse primer	CAACACTATTAGCATAAGCAGTTGT	[27]
SARS-CoV-2 probe	FAM-CAGGTGGAACCTCATCAGGAGATGC-BHQ	[27]
TBEV forward primer	GGGCGGTTCTTGTTCTCC	[28]
TBEV reverse primer	ACACATCACCTCCTTGTCAGACT	[28]
TBEV probe	FAM-TGAGCCACCATCACCCAGACACA-BHQ	[28]
WNV forward primer	TCAGCGATCTCTCCACCAAAG	[29]
WNV reverse primer	GGGTCAGCACGTTTGTCATTG	[29]
WNV probe	FAM-TGCCCGACCATGGGAGAAGCTC-BHQ	[29]
DENV forward primer	ATTAGAGAGCAGATCTCTG	[30]
DENV reverse primer	TGACACGCGGTTTC	[30]
DENV probe	FAM-TCAATATGCTGAAACGCG-BHQ	[30]
CHIKV forward primer	AAAGGGCAAACTCAGCTTCAC	[31]
CHIKV reverse primer	GCCTGGGCTCATCGTTATTC	[31]
CHIKV probe	FAM-CGCTGTGATACAGTGGTTTCGTGTG-TAMRA	[31]

## Data Availability

Not applicable.

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
