# Peer review of "A Syntenin Inhibitor Blocks Endosomal Entry of SARS-CoV-2 and a Panel of RNA Viruses"

_viruses, 2022, doi:10.3390/v14102202_

Round 1

Reviewer 1 Report

Authors of this manuscript tested the antiviral effect of the previously described syntenin inhibitor KSL-128114 against SARS-CoV2 and other enveloped viruses. They demonstrated an antiviral activity at the virus at the entry level by blocking endosomal trafficking.  It was suggested that the inhibitory activity is related to the systenin-dependent endocytic trafficking. The manuscript is straightforward and generally clear, below some more specific comments:

Minor Concerns and corrections

·      Introduction is very concise and lacks an explanation of the function of E protein in virus entry and of Nsp11.

·      Need to clarify KSL peptide as ‘cell penetrating’ since all the effects described could depend on vesicle trafficking.

·      On line 31 indicate the complete name of the term TMPRSS2.

·      Line 62-63 … block the interactions between synthenin’s role … not clear, modify.

·      Paragraphs from lines 54-59 should include a reference.

·      The term Nsp11 and Nsp12 should be written uniformly in the text.

·      Line 211 and 212, correct NSP3 in the text (Nsp11), results mentioned in this line correspond to the figure 1B instead of 1A.

·      The scheme of time of addition studies should be aligned with Figure D and the authors need to include a description of the experiment in the methodology session.

·      Please, change h to hours in the text.

·      Explain the term PM and total in the legend of figure 3.

·      Figure 3B black squares should be ACE2?

·      Line 347 corrects the term corona virus to coronavirus

Major concerns

·      In Figure 1 authors showed that systemin is able to bind Nsp11 and, to a lesser extent, E protein of SARS-CoV2. This interaction suggests these could be targets, although this evidence was poorly explored in the manuscript. Would there be a role for these viral factors in the mechanism proposed of entry inhibition? Is it only Syndecan the target, or both? Is it endosomal escape?

·      In figure 2C to understand if there is a reduction of viral particles, authors need to include an infectivity assay, using plaque assay to quantify virus viable particles to support the antiviral activity of KSL-128114. Can the authors explain why didn’t they evaluate the effect of inhibitor at time 0 in the time addition study?

·      To evaluate inhibitor activity against SARS-CoV2 in Vero E6 cells and Calu-3, the authors used an MOI of 0.05 at 16 hours infectious.  Explain why choosing a lower MOI, and how is the level of infection in these cells at this time point.  Are the cells show a sufficiently high level of infection to perform the antiviral assay?

·      In Figure 3C the author quantified the total of ACE2 in HEK293T-hACE2 cells. Why are the total levels of ACE2 in treated cells higher compared to untreated? In addition, the authors should include the quantification of the levels of ACE2 in Vero E cells, which was the cell model where the activity of the inhibitor was evaluated. Why are the authors evaluating the levels of ACE2 in HEK293T-hACE2 cells?

·      The authors demonstrated that KSL-128124 has no inhibitory effect in Calu-3 infected SARS-CoV2, although this result was poorly explored. A little more work should be done in the discussion on to entry inhibitors of SARS-CoV2 by looking at similar examples in the literature.

·      Finally, the authors concluded the inhibitor KSL-1281124 is active against DENV, WNV, TBEV, and CHIKV virus with IC50 30uM. Based on Figure 4 the activity for the mentioned viruses was higher compared to SARS-CoV2 activity (compare to figure 2B for example. Furthermore, the authors need to include TOA experiments and infectious assay for DENV, WNV, TBEV, and CHIKV to observe if there is reduction in progeny virus and if inhibition is similarly at the entry level.

Author Response

Reviewer 1

Minor Concerns and corrections

Comment 1:

  • Introduction is very concise and lacks an explanation of the function of E protein in virus entry and of Nsp11.

Reply:

We agree with the reviewer and have added the following information to the introduction:

“In addition to its role in assembly, budding and virulence, the E protein serves an important role during SARS-CoV-2 entry, as the E protein fuses with the endosomal membrane or plasma membrane in order to release the viral RNA into the cytoplasm [17,18]”

Lines 53-55

In addition to the reported PDZ binding motifs, there is a putative motif in the C-terminal region of the short intrinsically disordered NSP11 (full sequence SADAQSFLNGFAV-COO-). NSP11 becomes the N-terminus of NSP12 in the ribosomal frameshift of ORF1b. However, the function of NSP11 is not fully understood.

Lines 58-62

Comment 2:

  • Need to clarify KSL peptide as ‘cell penetrating’ since all the effects described could depend on vesicle trafficking.

Reply:

This is a very important and valid point, we have clarified this in introduction and in few places throughout the text.

Comment 3:

  • On line 31 indicate the complete name of the term TMPRSS2.

Reply:

We agree and have added the full name transmembrane serine protease 2 (TMPRSS2) line: 31

Comment 4:

  • Line 62-63 … block the interactions between synthenin’s role … not clear, modify.

Reply:

We agree and have clarified it as indicated below:

We reasoned that the KSL-128114 inhibitor could potentially be used to affect syntenin’s role in receptor trafficking, as well as potential interactions between syntenin and the SARS-CoV-2 proteins, which could affect viral infection.

Line 66-68

Comment 5:

  • Paragraphs from lines 54-59 should include a reference.

Reply:

Correct, we have now added a reference.

 Line 62

Comment 6:

  • The term Nsp11 and Nsp12 should be written uniformly in the text.

Reply:

We agree, and have updated to NSP11, NSP12 throughout the text and figures

Comment 7:

  • Line 211 and 212, correct NSP3 in the text (Nsp11), results mentioned in this line correspond to the figure 1B instead of 1A.

Reply:

We thank the reviewer for pointing out the mistake and have corrected this line 233.

Comment 8:

  • The scheme of time of addition studies should be aligned with Figure D and the authors need to include a description of the experiment in the methodology session.

Reply:

The figure was modified as the reviewer requested. We also added a paragraph to the material and methods to further clarify the experimental setup.

2.7. Time of addition assay

VeroE6 cells were treated with 30 μM KSL-128114 according to the following setup, (I, “-2”): cells were treated with 30 μM KSL-128114 for 2 hours at 37˚C and 5 % CO2 then medium containing peptide was removed and cell were infected with SARS-CoV-2 (MOI: 0.05) for 1 hour 37˚C and 5 % CO2, then inoculum was replaced with fresh medium con-taining 30 μM KSL-128114 and cells were incubated at 37˚C and 5 % CO2. (II, “1”): Cells were infected with SARS CoV-2 (MOI:0.05) for 1 hour 37˚C and 5 % CO2 then inoculum was replaced with fresh medium containing 30 μM KSL-128114 and incubated at 37˚C and 5 % CO2. (III, “3”) Cells were infected with SARS CoV-2 (MOI:0.05) for 1 hour at 37˚C and 5 % CO2then inoculum was replaced with fresh medium, after 2 hours medium was replaced with fresh medium containing 30 μM KSL-128114, and cells were incubated at 37˚C and 5 % CO2. After 16 hours of infection cells were fixed using 4 % formaldehyde and permeabilized in 0.5 % triton-X 100, 20 mM glycine in PBS. Infected cells were detect-ed using primary monoclonal rabbit antibodies directed against SARS-CoV-2 nucleocap-sid (Sino Biological Inc., 40143-R001), and conjugated secondary antibodies anti-rabbit Alexa555 (1:500, Thermo Fisher Scientific). Nuclei were counterstained with DAPI.

Lines 176-191

Comment 9:

  • Please, change h to hours in the text.

Reply:

This have been changed throughout the text

Comment 10:

  • Explain the term PM and total in the legend of figure 3.

Reply:

We thank the reviewer for pointing out the missing explanation. We have now clarified it in the figure legend as requested by the reviewer:

“Detection of ACE2; cells were permeabilized in order to detect ACE2 on the cellular surface as well as inside the cell “total”, or  cells were not permeabilized in order to detect ACE2 on the cellular surface “PM” (plasma membrane).”

Comment 11:

  • Figure 3B black squares should be ACE2?

Reply:

It should indeed be syntenin, however the comment made it clear to us that the figure needed to be clarified. We have now clarified the figure so it is easier to understand, by adding headings to the figure (KSL-128114 to A, and ACE2 to figure B)

Comment 12:

  • Line 347 corrects the term corona virus to coronavirus

Reply:

Done.

Major concerns

Comment 13:

  • In Figure 1 authors showed that systemin is able to bind Nsp11 and, to a lesser extent, E protein of SARS-CoV2. This interaction suggests these could be targets, although this evidence was poorly explored in the manuscript. Would there be a role for these viral factors in the mechanism proposed of entry inhibition? Is it only Syndecan the target, or both? Is it endosomal escape?

Reply:

We thank the reviewer for pointing out that the topic  deserved further discussion. We added the following to the discussion:

“ Finally, we confirmed that syntenin can interact with low affinity with the E protein of SARS-CoV-2, a protein that is involved in membrane fusion at the endosomal membrane [17]. If the interaction between syntenin and the E protein is needed to facilitate viral entry, the inhibitor could also contribute inhibition of viral entry by disrupting this interaction.

If KSL-128114 blocks trafficking of syndecans, CD63, other syntenin cargos, or direct interactions between syntenin and with viral proteins to confer the antiviral effect remain to be elucidated, and may vary from virus to virus.

 Lines 404 - 411

Comment 14:

  • In figure 2C to understand if there is a reduction of viral particles, authors need to include an infectivity assay, using plaque assay to quantify virus viable particles to support the antiviral activity of KSL-128114. Can the authors explain why didn’t they evaluate the effect of inhibitor at time 0 in the time addition study?

Reply:

The reviewer is correct that a plaque assay or focus forming assay is the golden standard for detecting infectious virus in the supernatant. However, in this case we made the decided that qPCR is the better choice, the reason for this is that there might be residual KSL-128114 left in the supernatant which could have an impact on downstream plaque assay (but not on qPCR).

Comment 15:

  • To evaluate inhibitor activity against SARS-CoV2 in Vero E6 cells and Calu-3, the authors used an MOI of 0.05 at 16 hours infectious. Explain why choosing a lower MOI, and how is the level of infection in these cells at this time point.  Are the cells show a sufficiently high level of infection to perform the antiviral assay?

Reply:

An MOI of 0.05 should give 5 % cells infected initially, and the replication cycle of SARS-CoV-2 is very fast with progeny virus coming out already at ~8 hours post infection. The MOI and incubation time has carefully been optimized for this assay so that we can detect both a positive or negative effect on viral infection (the infection of the control cells are 50 – 60 %).

Comment 16:

  • In Figure 3C the author quantified the total of ACE2 in HEK293T-hACE2 cells. Why are the total levels of ACE2 in treated cells higher compared to untreated? In addition, the authors should include the quantification of the levels of ACE2 in Vero E cells, which was the cell model where the activity of the inhibitor was evaluated. Why are the authors evaluating the levels of ACE2 in HEK293T-hACE2 cells?

Reply:

We agree with the reviewer’s comment that it would have been relevant to quantify the ACE2 levels in Vero E cells. However, the two ACE2 antibodies tested (ACE2, Novus bio, NBP2-67692, and ACE2, Abcam, ab15348 ) did not give a clear signal for ACE2 in Vero E cells. Thus, we resorted to using HEK293T-hACE2 cells.

We have added the following clarification to the result section:

”However, we found that the inhibitor increased the total amount of ACE2 expression, the most likely explanation is related to off-target effects of the inhibitor on a panel of other PDZ domains [5,23].”

Line 281-283

Comment 17:

  • The authors demonstrated that KSL-128124 has no inhibitory effect in Calu-3 infected SARS-CoV2, although this result was poorly explored. A little more work should be done in the discussion on to entry inhibitors of SARS-CoV2 by looking at similar examples in the literature.

Reply:

We agree with the reviewer that the topic deserved further elaboration. We have added the following to the discussion:

“In that sense, KSL-128114 showed similar inhibition patterns as chloroquine which strongly inhibited SARS-CoV-2 infection in VeroE6 cells but not in Calu-3 cells. Chloroquine acts by increasing the pH within the endosomes and thus inhibits viruses that depend on low pH for their entry [35]. Previous studies have indeed showed similar inhibition patterns in VeroE6 and Calu-3 cells using chloroquine [33,36], indicating that the entry mechanism in VeroE6 cells are dependent on the endosomal pathway.”

Line 375-381

Comment 18:

  • Finally, the authors concluded the inhibitor KSL-1281124 is active against DENV, WNV, TBEV, and CHIKV virus with IC50 30uM. Based on Figure 4 the activity for the mentioned viruses was higher compared to SARS-CoV2 activity (compare to figure 2B for example. Furthermore, the authors need to include TOA experiments and infectious assay for DENV, WNV, TBEV, and CHIKV to observe if there is reduction in progeny virus and if inhibition is similarly at the entry level.

Reply:

We agree with the reviewer that these are interesting experiments, but find them better suited for a follow up studies as SARS CoV-2 was the main focus of this study.

Reviewer 2 Report

Line 74-75 the authors state: encoding the 6-His-GST fusion.  Which fusion proteins? Specify.

Line 84 the authors state that the proteins were confirmed by SDS-PAGE but there isn’t any data about the purification process or the isolated proteins size and purity.

Unlabeled ---- peptides were dissolved in 50 mM potassium phosphate, pH 7.5.

FITC-labeled peptide in 50 mM potassium phosphate, pH 7.5. ? What is the %DMSO in labeled peptides?

Figure 3 C and D: what does it mean PM?

Harmonize all the plots (Font, subtitles)

Discussion/Concluions

Authors claimed that the KSL-128114 inhibit infection of a broad range of enveloped RNA viruses, most likely by blocking the entry through the endosomal pathway. 

I would suggest an additional experiment to prove this statement, since it is not clear from the experiments performed.

Authors could use KSL-128114 treatment and use specific endosomal markers to colocalize them.

Author Response

Reviewer 2

Comment 1:

Line 74-75 the authors state: encoding the 6-His-GST fusion.  Which fusion proteins? Specify.

Reply:

We thank the reviewer for pointing out the lack of information. We have now specified it in the text “(human Syntenin PDZ1-2 (amino acids 111-213), MPP5 PDZ (amino acids 238-336) and SNX2 PDZ (amino acids 740-141); synthetic genes obtained from GeneScript))”.

Line 78-81. 

Comment 2:

Line 84 the authors state that the proteins were confirmed by SDS-PAGE but there isn’t any data about the purification process or the isolated proteins size and purity.

 Reply:

We thank the reviewer for pointing out the lack of information, which allowed us to identify a mistake in the method section, which we have now corrected. The updated method section reads:

“Proteins were batch purified from the supernatant using Ni Sepharose® Excel (Cytiva) using the manufacturer’s recommended buffers. The supernatant was mixed with the matrix and unbound fraction was washed out with wash buffer (20 mM NaPi, 0.5 M NaCl, 30 mM imidazole, pH 7.4). The bound protein was cleaved with His-tagged 3C protease (in 20 mM NaPi, 0.5 M NaCl, pH 7.4) at 4˚C for 16 hours. The proteolytically released PDZ domains were obtained from the matrix by addition of buffer. The protein size and purity were confirmed through SDS-PAGE. Purified proteins were dialyzed into 50 mM potassium phosphate buffer, pH 7.5, for 16 hours.”

Line 89-96.

We routinely analyze all purified proteins by SDS-PAGE analysis, but we do not usually do this for publication purposes. That being said, we keep of course a record of the SDS-PAGE gels. We have added the gels of the purified PDZ domains below as requested by the reviewer. The sizes of the PDZ domains are 9.5k Da MPP5 PDZ, 18. 2kDa for Syntenin PDZ1-2 and 11.2 kDa for SNX27 PDZ after removal of the GST-tag.

Unlabeled ---- peptides were dissolved in 50 mM potassium phosphate, pH 7.5.

Figure 1. SDS-PAGE analysis of purified proteins. A. MPP5 PDZ. B. SNZ27 PDZ before and after proteolytic removal of the GST-tag. C. Syntenin PDZ1-2 together with un unrelated protein. The ladder was Biorad Precision Plus Protein™ Unstained Protein Standards 1 ml #1610363

Comment 3:

FITC-labeled peptide in 50 mM potassium phosphate, pH 7.5. ? What is the %DMSO in labeled peptides?

Reply:

The lyophilized FITC-labeled peptides were dissolved in DMSO at an average concentration of 6 mM. The peptides were then dilute the peptides in buffer into a final FITC-peptide concentration of 10 nM. The DMSO concentration during the affinity measurements is thus less than 0.1 promille. 

Comment 4:

Figure 3 C and D: what does it mean PM?

Reply:

We agree that this was not clear and have added in order to clarify

“Detection of ACE2; cells were permeabilized in order to detect ACE2 on the cellular surface as well as inside the cell “total”, or  cells were not permeabilized in order to detect ACE2 on the cellular surface “PM” (plasma membrane).”

Comment 5:

Harmonize all the plots (Font, subtitles)

Reply:

We thank the reviewer for the comment, we have further streamlined the figures which has improved the manuscript.

 Comment 6:

Discussion/Conclusions

Authors claimed that the KSL-128114 inhibit infection of a broad range of enveloped RNA viruses, most likely by blocking the entry through the endosomal pathway. 

I would suggest an additional experiment to prove this statement, since it is not clear from the experiments performed.

Authors could use KSL-128114 treatment and use specific endosomal markers to colocalize them.

Reply:

We agree with the comment, but respectfully find the detailed analysis with endosomal markers is a valid topic for a follow up study. Following the suggestion we added the following clarification regarding syntenins interaction with endosomal markers, and its potential role in inhibition of viral entry.

”The KSL-128114 inhibitor has previously been shown to have a negative effect on syntenin-dependent endosomal budding through binding to syntenin PDZ1 and has been shown to block an interaction between syntenin and RAB5 [23], a key factor in regulating early endocytosis [43,44]. The impairment of the PDZ1 binding pocket has further previously been shown to lead the co-accumulation of syntenin and syndecan in a recycling compartment [8].

Line 381-386

“Furthermore, KSL128114 was shown to disrupt the interaction between Rab5 and syntenin [23], and Rab5 is needed for DENV and WNV entry in to cells [45], which may suggest an interesting mechanism for KSL-128114 inhibition of viral entry. “

Line 400-403

Reviewer 3 Report

Manuscript entitled “A syntenin inhibitor blocks endosomal entry of SARS-CoV-2 and a panel of RNA viruses” by Lindqvis et al described highly potent and metabolically stable peptide inhibitor, KSL-128114 that binds to the PDZ1 domain of syntenin inhibits SARS-CoV-2 infection by blocking the endosomal entry of the virus. In addition, the Authors also found that the inhibitor inhibits chikungunya and flavivirus infection, which are completely dependent on receptor mediated endocytosis for their entry. This manuscript has few merits and demerits. Authors have synthesized and purified the peptide, KSL-128114 followed by testing them in VeroE6, Calu-3 and VeroB4 cells. The drug was tested for binding efficiency and entry inhibition using qPCR. In addition, Authors have also used FP-based assay to determine the binding of KSL-128114 to syntenin PDZ1-2 or SNX27 PDZ. For SARS-CoV-2 infection, they have used a patient isolate, SARS-CoV-2/01/human/2020/SWE. Based on the qPCR, Authors found that KSL-128114 did not block the binding of the SARS-CoV-2 virus to the VeroE6 cells, but rather the post-endocytic entry of the virus to the cytoplasm. These results are quite interesting and demonstrate the importance of syntenin for infection of a wide-range of different RNA viruses. Demerits of the study is they failed to demonstrate protein-protein interaction using western blot. In addition, they have not demonstrated results for purification of inhibitors, but it’s not important as the study is focused on the inhibitory effect of KSL-128114 on SARS-CoV2 and other viral infections. Overall, the study has interesting results for the readers. Hence, I would recommend the manuscript for publication in the journal.

Author Response

No specific comments were raised by this reviewer.

Round 2

Reviewer 1 Report

authors addressed all questions and provided and improved manuscript

Reviewer 2 Report

Thank you for all corrections and comments.